# A Decentralized Study Setup Enables to Quantify the Effect of Polymerization and Linkage of α-Glucans on Post-Prandial Glucose Response

**DOI:** 10.3390/nu14051123

**Published:** 2022-03-07

**Authors:** Frederik Delodder, Andreas Rytz, Fabien Foltzer, Lisa Lamothe, Carmine d’Urzo, Ludivine Feraille-Naze, Julia Mauger, Justine Morlet, Nathalie Piccardi, Lionel Philippe, François Caijo, Jeroen Schmitt, Sara Colombo Mottaz

**Affiliations:** 1Nestlé Research Center, 1000 Lausanne, Switzerland; frederik.delodder@rdls.nestle.com (F.D.); fabien.foltzer@gmail.com (F.F.); lisa.lamothe@rdls.nestle.com (L.L.); ludivine.feraille-naze@rdls.nestle.com (L.F.-N.); julia.mauger@rdls.nestle.com (J.M.); justine.morlet@rdls.nestle.com (J.M.); nathalie.piccardi@rdls.nestle.com (N.P.); lionel.philippe@rdls.nestle.com (L.P.); frcaijo@gmail.com (F.C.); jeroen.schmitt@gmail.com (J.S.); sara.colombomottaz@rd.nestle.com (S.C.M.); 2Nestlé Health Science, 1000 Lausanne, Switzerland; carmine.durzo@nestle.com; 3CHU Toulouse, 31000 Toulouse, France; 4Singapore Institute of Food and Biotechnology Innovation (SIFBI), Agency for Science, Technology and Research (A*STAR), Singapore 138671, Singapore

**Keywords:** post-prandial glucose response (PPGR), α-glucans, polymerization, decentralized study design

## Abstract

The complexity of the carbohydrate structure is associated with post-prandial glucose response and diverse health benefits. The aim of this study was to determine whether, thanks to the usage of minimally invasive glucose monitors, it was possible to evaluate, in a decentralized study setup, the post-prandial glycemic response (PPGR) of α-glucans differing systematically in their degree of polymerization (DP 3 vs. DP 60) and in their linkage structure (dextrin vs. dextran). Ten healthy subjects completed a double-blind, randomized, decentralized crossover trial, testing at home, in real life conditions, four self-prepared test beverages consisting of 25 g α-glucan dissolved in 300 mL water. The incremental area under the curve of the 120 min PPGR (2h-iAUC) was the highest for Dextrin DP 3 (163 ± 27 mmol/L*min), followed by Dextrin DP 60 (−25%, *p* = 0.208), Dextran DP 60 (−59%, *p* = 0.002), and non-fully caloric Resistant Dextrin (−68%, *p* = 0.002). These results show that a fully decentralized crossover study can be successfully used to assess the influence of both polymerization and structure of α-glucans on PPGR.

## 1. Introduction

Carbohydrates play a major role in human nutrition, supplying close to half of all calories and facilitating numerous metabolic functions. Free sugars should account for less than 10% of total energy as recommended by the World Health Organization [1]. Consequently, a large part of the glucose load of most diets is highly imputable to polysaccharides and more specifically to starches, glucose syrups and maltodextrins that are omnipresent in most diets. These are all α-glucans, which are polysaccharides of D-glucose monomers linked with glycosidic bonds of the alpha form. Since foods and beverages with low post-prandial glucose response, low glycemic index (GI), and low glycemic load (GL) are considered beneficial for the prevention of type-2 diabetes [2] and cardio-vascular disease [3] as well as for the management of diabetes [4], it is necessary to pay special attention to the glycemic power of these α-glucans, although other macro- and micronutrients further modulate the glucose response [5].

Both starches and maltodextrins are readily digested in the gastrointestinal tract by salivary and pancreatic α-amylase, leading to rapid increases in blood glucose. However, there is growing interest in using complex carbohydrates, which are slowly digested throughout the small intestine, resulting in a slow and prolonged provision of glucose into the blood over time, resulting in a low glycemic response [6]. These types of carbohydrates that result in a slow rise and fall of blood glucose are referred to as slowly digestible carbohydrates. Enzymatic modifications of starch structure to modulate its digestibility have been extensively studied. Modifications to alter the digestibility include increasing chain length (polymerization), cyclization, altering linkage profiles, and/or increasing branching [7].

Following this logic, the objective of the present study was to compare the glycemic response of α-glucans differing systematically in their degree of polymerization (DP 3 vs. DP 60) and structure (linear dextrin vs. branched dextran). The comparisons were performed in healthy adults using minimally invasive glucose monitors to assess whether it was possible to obtain conclusive results out of a decentralized crossover study.

The decentralized setup included the use of social media for adverts, recruitment and enrolment, online initiation, at-home consent monitored through Visio conference, remote real-time safety overseeing, direct-to-subject product and materiel delivery, self-placement and removal of a minimally invasive continuous glucose monitor, self-product preparation and consumption, and an electronic diary for product tolerance. Compliance to procedures and product intake was assessed via electronic patient-reported outcome (ePRO). The decentralized setup potentially reduces the burden on both the subjects and investigational site while providing real-world evidence [8] and rapidly actionable results, even under special conditions such as COVID-19 pandemics [9].

## 2. Materials and Methods

### 2.1. Samples

The study compared three α-glucans that systematically varied the structure of glycosidic bonds linking glucose units (α-1,4 vs. α-1,6) and the degree of polymerization (3 to 60). A low glycemic control was added that is composed of 70% Resistant Dextrin and 30% Dextrin; its average molecular weight of MW = 990 Da can be considered equivalent to a degree of polymerization of DP = 5, since MW = 180 Da for glucose (Table 1). These powders were then dissolved in still, non-aromatized water (25 g/300 mL). The first three samples provided 100 kcal, while the control provided only 30 kcal.

### 2.2. In-Vivo Study

Ten healthy subjects (eight women, two men) were recruited with age mean = 31.8 y and SD = 8.5 y; BMI mean = 21.5 and SD = 1.9 kg m^−2^; and fasting glucose mean = 4.7 and SD = 0.58 mmol/L. One day after self-placement of the Abbott Freestyle Libre [10], participants started the testing of the four products, one on each of four consecutive days, after 12 h overnight fasting. Fasting interstitial glucose flash readings were obtained 5 min before (T-5) and at time of beverage intake (T0). These two values were averaged to serve as the baseline glucose value. After the baseline value was obtained, subjects consumed, within a few minutes, one of the four test beverages, and the continuous glucose monitoring delivered a value every 15 min. Participants were then asked to assess their gastro-intestinal tolerance according to six dimensions (abdominal discomfort, decreased appetite, gastric reflux, nausea, diarrhea, headache) using a visual analogue scale. Due to the nature of the products (i.e., almost taste-neutral ingredients), the focus was kept on gastro-intestinal tolerance without introducing other dimensions such as preference.

The study followed a double-blind, randomized, decentralized cross-over design, with subjects being randomly allocated to a sequence of the four tested conditions using a Williams Latin square design that counterbalanced position and first order carry-over effects [11]. In this setup, each subject tested each product once.

Since no comparable decentralized study had been performed earlier, the design was adaptive in terms of sample size, allowing for a maximum of 30 subjects, with interim analyses foreseen after 10 and 20 subjects and the possibility to stop for futility (i.e., no difference between positive and negative controls Dextrin DP3 and Resistant Dextrin) or for success of the primary objective (i.e., significant difference between Dextrin DP3 and Dextran DP 60). The study was stopped for success after 10 subjects, which is the minimal number required for assessing post-prandial glucose response [12].

Participants signed an informed consent form as per local regulations and the study protocol was reviewed and approved by the Ethics Committee of Canton de Vaud (Lausanne, Switzerland, study reference 2019-01431) and registered (clinicaltrials.gov, NCT05266690).

### 2.3. Technical Aspects of Decentralization

A major challenge of this decentralized study was the management of material and data flows, while ensuring compliance with Data Protection Laws and Human Research Acts. To do so, the following technical means have been implemented (Table 2). It is important to note that at the time of the study, the full e-consent process was not allowed in Switzerland; therefore, Visio conferencing was used for the information session to ensure that all questions could be addressed. The consent document (two copies) was sent by mail to the home of the participants. The signature of the participants was performed live during a Visio conference with the designee of the medical responsible of the study in attendance. Then, the two original copies were sent back to the site for the medical responsible/designee completion. Once enrolled, subjects received all products and materials through the Swiss National mailing service (La Poste). Medidata Patient Cloud was used to collect electronic patient-reported outcomes (ePRO), allowing safety oversight, and monitoring of product tolerance and compliance.

### 2.4. Data Analyses

The primary endpoint of this study was the 2 h incremental area under the curve (2h-iAUC) of post-prandial glucose response. This 2h-iAUC was estimated using the trapezoid method on each individual curve. Secondary endpoints derived from the post-prandial glucose response were the maximal incremental glucose value (iCmax), the time to reach this value (Tmax), and all cross-sectional timepoints, every 15 min between 0 and 120 min. The mean glucose response curves are shown in a graph using the mean and standard error (SE) at each cross-sectional time-point (Figure 1).

Endpoints derived from glucose curves are tabulated using Mean ± SE and *p*-values associated with the paired *t*-test vs. the Dextrin DP 3, with a two-sided 5% significance level (Table 3). Since the primary objective was the comparison of Dextrin DP 3 vs. Dextran DP 60, no correction for multiplicity was applied. A sensitivity analysis was performed using a mixed model to account for potential systematic position or carry-over effects [13]. Since none of these effects were close to reaching statistical significance, these analyses are not further presented.

The gastrointestinal tolerance assessment appeared to be strongly zero-inflated (i.e., >73% observed values were zero). Consequently, data are discussed as distributions among three classes: zero symptom (value = 0), mild symptom (0 < value ≤ 20), higher symptoms (value > 20). The values of the latter class are tabulated (Table 4).

## 3. Results

### 3.1. Post-Prandial Glucose Response

The average 2h-PPGR curves show that the fasting baseline value is 4.7 ± 0.58 mmol/L (Mean ± SE, N = 10) and that all test beverages peak around 45 min before coming back to baseline at around 90 min (Figure 1). The mean 2h-PPGR curves appear to be highest, between 0 and 90 min, for Dextrin DP 3, followed by Dextrin DP60 and Dextran DP60 that is close to the Resistant Dextrin control. The shapes of these curves show that the main difference concerns the height of the glucose excursion, much more than a shift in time.

These curves translate into the highest 2h-iAUC for Dextrin DP 3 (163 ± 27 mmol/L*min), followed by Dextrin DP 60 (25% decrease, *p* = 0.208), Dextran DP 60 (59% decrease, *p*-value = 0.002) and Resistant Dextrin (68% decrease, *p*-value = 0.002) (Table 3).

In terms of incremental Cmax, this translates into the highest iCmax for Dextrin DP 3 (3.5 ± 0.36 mmol/L), followed by Dextrin DP 60 (28% decrease, *p* = 0.019), Dextran DP 60 (62% decrease, *p*-value < 0.001), and Resistant Dextrin (66% decrease, *p*-value < 0.001).

The glucose peak appears on average after, respectively, 46 ± 2.7 min for Dextrin DP 3 and slightly earlier for the three other products, without reaching statistical significance.

### 3.2. Gastrointestinal Tolerance

The gastrointestinal tolerance was assessed using a visual analogue scale (score 0–100) for six different symptoms. Overall, the scores were very low, indicating good tolerance for all tested products, with 95% scores lower than 20 and 73% even being zero. The number of subjects out of 10 scoring higher than 20 is tabulated (Table 4). These results suggest that, in healthy subjects, the gastrointestinal tolerance to all tested products was very high.

## 4. Discussion

Carbohydrates are a major constituent of our diet with a minimal recommended dietary allowance of 130 g/day [14]. It is therefore important to ensure their quality, with a focus on reducing their glycemic response to improve cardiometabolic conditions [15]. The objective of this research was to quantify, in healthy subjects, the effect of polymerization and linkage of α-glucans on the post-prandial glucose response using a decentralized subject-centric study setup.

This decentralized study shows that ten subjects were sufficient to significantly differentiate Dextrin DP 3 from Resistant Dextrin as well as from other fully digestible α-glucans. Alternative α-glucans with a higher degree of polymerization (DP 60 vs. DP 3) and more branched structure (α-1,6 instead of α-1,4 glycosidic bonds) led to significantly lower post-prandial glycemic responses.

Dextrin DP 60 reduced the post-prandial glucose response by 25% as compared to Dextrin DP 3 (i.e., 123 ± 15 vs. 163 ± 27 mmol/L*min). Although this decrease is remarkable, and even statistically significant for the incremental raise in glucose, it has a limited practical impact since α-glucans commonly used in the food industry are glucose syrups (DP ≤ 5) or maltodextrins with DP ≤ 30 that have lower degrees of polymerization than DP 60 [16]. Consequently, it is not surprising that, independent of their degree of polymerization, the frequent consumption of maltodextrins should be judged with caution [17]. The tested DP 60 was used to have a direct comparison with ingredients rich in α-1,6 glycosidic bonds, for which dextran DP 60 was chosen as a relevant model.

Dextran DP 60 reduced the post-prandial glucose response by 46% as compared to Dextrin DP 60 (i.e., 67 ± 14 vs. 123 ± 15 mmol/L*min). This shows that, at a comparable polymerization, the structure plays a key role, with α-1,6 glycosidic bonds being close to half as glycemic as α-1,4 glycosidic bonds. It is further interesting to notice that Dextran DP 60 has a glucose response that is almost superposed with the Resistant Dextrin control, which lead to an almost 70% decrease in 2h-iAUC as compared to Dextrin DP 3. This 70% reduction was expected knowing that the Resistant Dextrin control was composed of 70 resistant and 30% digestible dextrin [18] The fact that Dextran 60, which is fully digestible, reduces 2h-iAUC almost as much as this control is consistent with results obtained with pullulan, another fully digestible polysaccharide with a structure featuring both α-1,4 and α-1,6 glycosidic bonds. For 50 g pullulan, a reduction of 50% in 3h-iAUC vs. 50 g maltodextrin has been reported for an average molecular weight > 100 kDa, equivalent to DP > 555 [19]. For lower molecular weight (6300 Da, equivalent to DP 35), 50 g pullulan did not reduce the glucose response, confirming our finding that, in conjunction with the structure, molecular weight plays an important role [20]. These results confirm that structural factors modulate the digestion speed of fully digestible carbohydrates [21], and govern starch digestion and glycemic responses [22].

Thanks to this study, we could show that the decentralized study setup is fully suitable to detect clinically relevant modulations of post-prandial glucose response with only ten subjects, which is the minimal number defined for conventional centralized studies [12]. This shows that self-placing of a minimally invasive continuous glucose monitoring device, self-preparation and administration of test products, and subject compliance to procedures can be reached in such a setup. Interestingly, when analyzing the continuous glucose trace two hours before the test product intake, the traces are constant, demonstrating high compliance to fasting and no other variations, potentially due to stressful situations to reach the central location, as this has been observed previously in some conventional studies performed with continuous glucose monitoring. Those insights show an advantage in term of stress-bias with the decentralized approach versus the traditional centralized one. Our decentralized trial was able to offer a subject-centric approach, allowing study participants to run the study with total flexibility and avoiding the burden of a traditional set up such as fixed on-site visits and travel.

The shift of clinical trial activities closer to subjects has been enabled by a constellation of evolving technologies and services. Tools such as telemedicine, remote subject monitoring, and electronic clinical-outcome assessments allow investigators to maintain links to trial participants and enable more procedures to occur away from research sites without the need for in-person visits. From a societal aspect, decentralization broadens the trial access to reach a larger number and potentially a more diverse pool of subjects. The COVID-19 pandemic has catalyzed the adoption of decentralized clinical trials to ensure participants and staff safety while enabling ongoing study execution.

In conclusion, this study confirms that through the usage of minimally invasive glucose monitors within a decentralized crossover study, we can obtain high quality and conclusive results regarding the evidence that both the polymerization and structure of α-glucans strongly influence their post-prandial glucose response. This brings additional evidence that ingredient developments modulating these two parameters are essential to improve the quality of currently used α-glucans. Further studies using less extreme conditions (e.g., with DP between 3 and 15) and more diverse structure are required to support ingredient development. This decentralized study shows that such further assessments could be performed using a similar approach, which could have multiple advantages in terms of simplified operations, high data quality, and reduced subject burden.

## Figures and Tables

**Figure 1 nutrients-14-01123-f001:**
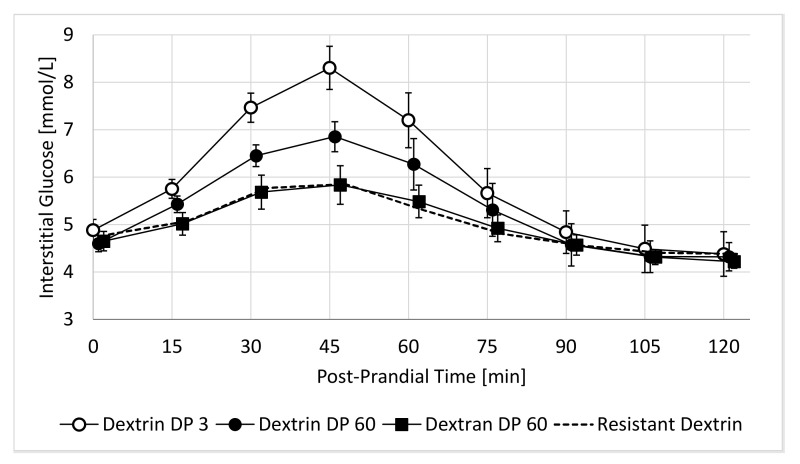
Average 2h-PPGR of the four test beverages featuring 25 g powder dissolved in 300 mL water. Data are shown as Mean ± SE at cross-sectional time-points every 15 min (N = 10 subjects).

**Table 1 nutrients-14-01123-t001:** Four samples with structure (glycosidic bonds), degree of polymerization (DP), and commercial name.

Name	Glycosidic Bonds	DP	Commercial Name
Dextrin DP 3 Dextrin DP 60	α-1,4 α-1,4	3 60	Roquette Glucidex 40 Roquette Glucidex 2
Dextran DP 60 Resistant Dextrin	α-1,6 -	60 ~5	Pharmacosmos Dextran 10 Promitor 70

**Table 2 nutrients-14-01123-t002:** Technical means implemented for each purpose of the decentralized study.

Purpose	Technical Means
Recruitment and enrolment At-home consent discussion Product and material delivery Continuous glucose monitoring Safety oversight Product tolerance Product compliance Procedure compliance	Internal social media (Workplace) Skype meeting National mailing service (La Poste) Abbott, Freestyle Libre (with offline reader) ePRO ePRO ePRO ePRO
Data reconciliation and aggregation	SAS^®^ Life Science Analytics Framework (LSAF)

**Table 3 nutrients-14-01123-t003:** Descriptive statistics (Mean ± SE, N = 10) for the four products featuring 25 g powder dissolved in 300 mL water. For pairwise comparisons vs. Dextrin DP 3 (paired *t*-test, two-sided), *p*-values are given in brackets.

Endpoint	Dextrin DP 3	Dextrin DP 60	Dextran DP60	Resistant Dextrin
2h-iAUC [mmol/L × min] iCmax [mmol/L] Tmax [min]	163 ± 27 3.5 ± 0.36 46 ± 2.7	123 ± 15 (*p* = 0.208) 2.5 ± 0.23 (*p* = 0.019) 43 ± 4.2 (*p* = 0.555)	67 ± 14 (*p* = 0.002) 1.3 ± 0.27 (*p* < 0.001) 38 ± 4.6 (*p* = 0.168)	52 ± 13 (*p* = 0.002) 1.2 ± 0.19 (*p* < 0.001) 38 ± 3.4 (*p* = 0.051)

**Table 4 nutrients-14-01123-t004:** Gastrointestinal tolerance: number of subjects out of 10 reporting symptoms with a score > 20 on a visual analogue scale (0–100).

Endpoint	Dextrin DP 3	Dextrin DP 60	Dextran DP 60	Resistant Dextrin
Abdominal discomfort Decreased appetite Gastric reflux Nausea Diarrhea Headache	0 0 0 0 1 0	1 0 0 1 0 4	0 0 0 0 1 1	0 0 1 1 2 0

## Data Availability

The data presented in this study are available on request from the corresponding author. The data are not publicly available due to privacy restrictions.

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
