# Peer review of "A Decentralized Study Setup Enables to Quantify the Effect of Polymerization and Linkage of α-Glucans on Post-Prandial Glucose Response"

_nutrients, 2022, doi:10.3390/nu14051123_

Round 1

Reviewer 1 Report

Although the title of the present manuscript suggests that the objective of the study was to investigate “the effect of polymerization and linkage of a-glucans on post-prandial glucose response”, it seems that the actual purpose was to test the usage of minimally invasive glucose monitors, in a decentralized study setup. Perhaps the title should be changed.

The study was designed to include 30 subjects, but was interrupted after 10 subjects, because conclusive results were obtained. Nevertheless, Methods and Results were not sufficiently clear to describe how many times each subjected received each preparation, and how the results varied in each subject, in different days (individual variation, different days). Furthermore, there are no data on taste or personal preference for the different preparations. These data should be included.

Author Response

Thank you very much for underlying so clearly the double objective of our work: it was both to quantify the effects of polymerization and linkage of a-glucans and to test the feasibility to perform such a quantification using a decentralized study setup. In order to better capture this double objective, we propose to add the words "setup enables" to the title, which now becomes: "A decentralized study setup enables to quantify the effect of polymerization and linkage of a-glucans on post-prandial glucose response".

In order to improve the clarity of the "Materials and Methods" section, we make it explicit that "each subject tested each product once" (line 96) and that "Due to the nature of the products (i.e., almost taste-neutral ingredients), the focus was kept on gastro-intestinal tolerance without introducing other dimensions such as preference" (line 90).

Reviewer 2 Report

This paper by Delodder and collaborators reports the results of a decentralized study to quantify the effect of polymerization and linkage of alpha-glucans on post-prandial glucose response.

The analyses were conducted by adopting decentralizated procedures and easy protocols involving individuals that in autonomy (at home) performed administration of the beverage and set up the analytical procedure. This represents an interesting aspect of the paper, i.e. there is the possibility to get to good data even if remote approaches are adopted.

Major points

None

Minor points

Please carefully check the papers list for the format. It should be uniform (compare e.g. the format of paper 2 or 3 to that of papers 6 or 7)

There are typos scattered throughout the manuscript (see e.g. line 184)

Author Response

Thank you very much for summarizing so clearly the double objective of our work. In order to underline this, and following the suggestions of another reviewer, we have slightly adapted the title that now becomes: "A decentralized study setup enables to quantify the effect of polymerization and linkage of a-glucans on post-prandial glucose response".

Following your suggestion, we revised the format of the bibliography. All journal titles are now spelled out completely and the punctuation is also made uniform.

We also revised the document for typos (e.g., "tan"-->"than", line 187 as suggested, but also lines 59, 75, 99, 101, 128, 155 and 226).